# Derivative Three-Dimensional Synchronous Fluorescence Analysis of Tear Fluid and Their Processing for the Diagnosis of Glaucoma

**DOI:** 10.3390/s22155534

**Published:** 2022-07-25

**Authors:** Katarína Dubayová, Kristína Krajčíková, Mária Mareková, Vladimíra Tomečková

**Affiliations:** Department of Medical and Clinical Biochemistry, Faculty of Medicine, Pavol Jozef Šafárik in Košice, Trieda SNP 1, 040 11 Košice, Slovakia; katarina.dubayova@upjs.sk (K.D.); kristina.ugrayova@gmail.com (K.K.); maria.marekova@upjs.sk (M.M.)

**Keywords:** tear fluid, glaucoma, second derivative 3D-synchronous spectra

## Abstract

Background: Sensitive and rapid diagnosis of the early stages of glaucoma from tear fluid is a great challenge for researchers. Methods: Tear fluid was analyzed using three-dimensional synchronous fluorescence spectroscopy (3D-SFS). Our previously published results briefly describe the main methods which applied the second derivative to a selected synchronous spectrum Δλ = 110 nm in distinguishing between healthy subjects (CTRL) and patients with glaucoma (POAG). Results: In this paper, a novel strategy was used to evaluate three-dimensional spectra from the tear fluid database of our patients. A series of synchronous excitation spectra were processed as a front view and presented as a single curve showcasing the overall fluorescence profile of the tear fluid. The second derivative spectrum provides two parameters that can enhance the distinction between CTRL and POAG tear fluid. Conclusions: Combining different types of 3D-SFS data can offer interesting and useful diagnostic tools and it can be used as input for machine learning and process automation.

## 1. Introduction

The composition of tear fluid has diagnostic potential, as it provides information about health and pathologic state of the eyes, but also the function of the whole body. Despite intensive research in the last few years, it is not used in practice yet. Tear fluid as sampling material is accessible, with a non-invasive and simple collection, and it shows lower variability of containing compounds compared to blood serum or plasma [1].

The tear fluid concentration of a large variety of salts and organic components (proteins, lipids, nucleic acids, mucins, metabolites) can be altered by any imbalance in the human organism, e.g., during inflammatory processes. Glaucoma represents a group of eye diseases that progressively damage the optic nerve and show multifactorial etiology. Most types of glaucoma develop without symptoms and, if left untreated, could lead to visual damage and blindness. A major factor is high intraocular pressure [1,2].

The attempts to find an accurate and sensitive biomarker are made by examining all three tear fluid layers: lipid, aqueous, and mucous. The aqueous layer is the most diverse tear film layer compared to the lipid and mucous layer. The source of biomarkers is crucial for specificity and reliability of diagnostic tests and treatment targets. The well-justified principle of a “sick eye in sick body” makes comprehensive tear fluid biomarker profiling highly relevant not only for the diagnosis of eye pathologies but also for prediction, prognosis, and personalized treatment of systemic diseases [3].

Metabolomic analysis of tear fluid has been used for diagnostic purposes in many studies in the case of ocular diseases which are characterized by loss of tear production, e.g., glaucoma [4], dry eye syndrome (DES) [5], and systemic diseases, such as diabetes mellitus [6], major depressive disorder [7], or even cancer [8,9]. For identifying changed content of tear fluid, researchers use various methods. Tear fluid is a complex analytical system with small volume that requires sensitive analysis. The use of an appropriate analytical method that characterizes its composition without separation procedures but is sensitive to changes caused by pathological conditions is crucial. Three-dimensional fluorescence analysis in the form of constant wavelength synchronous spectra (CW) shows enormous potential [2,6,7].

Tear fluid autofluorescence provides an intense imprint in the UV region that results from the presence of aromatic amino acids, mostly as components of proteins. Metabolites and derivatives of aromatic amino acids are also fluorescently active [2,10]. The synchronous spectrum can be sensitive to changes in the composition of the tear fluid at the metabolome level (fluorescent metabolites), but also at the protein level (aromatic amino acids). The change in structure affects the fluorescence in both intensity and wavelength shift. According to this assumption, synchronous fluorescence analysis can be used to distinguish between tear fluid from a glaucoma patient and from healthy subjects [2,10]. The use of 3D-SFS combined with arithmetic processing of the spectra such as second derivative (2D), enhances minor spectral features and allows more reliable identification of changes in a multifluorescent complex mixture [11]. Our previously published results [12] show a positive outcome when applying the second derivative to the selected synchronous spectrum Δλ = 110 nm in distinguishing between CTRL and POAG. In this work, the same 3D synchronous spectra of patients and controls were used to implement the idea of using second derivative spectra to identify glaucoma from tear fluid. A novel strategy has been proposed to evaluate the 3D-SFS. The work builds on a previously published pilot study [12].

## 2. Materials and Methods

### 2.1. Patients and Biological Material

In this article, synchronous excitation spectra of tear fluid from our database were used. The same 3D synchronous spectra of tear fluid of the patients and healthy subjects were used in our previously published pilot study [12] in which the tear fluid was collected by glass microcapillary at the Department of Ophthalmology, University Hospital Louis Pasteur in Košice from random patients (Table 1) with primary open-angle glaucoma (POAG, n = 13) and healthy subjects (CTRL, n = 8). The patients were treated according to the Declaration of Helsinki. They were informed about the objectives and risks of the study after which they signed the informed consent. The diagnosis of glaucoma and related diseases was based on the guidelines from the European Glaucoma Society (European Glaucoma Society Terminology and Guidelines for Glaucoma, 4th). 

Description of patients, procedures, and measurement conditions are described in detail in paper [12]. 

The study was approved by the ethical committee of Louis Pasteur University Hospital in Košice, (protocol code 2020/EK/06042 and 25 June 2020 date of approval) for studies involving tear fluid of humans.

### 2.2. Fluorescence Analysis

Tear fluid was stored at −80 °C until fluorescence analysis was carried out in quartz dark microcuvette on Perkin-Elmer Luminescence Spectrophotometer LS 55. The setting of the instrument’s excitation and emission slits was 5, 5 nm. The synchronous fluorescence spectra were analyzed in range from 200–600 nm with scan rate 1200 nm/min. The measured data were processed by WinLab Software, Buckinghamshire, United Kingdom. Synchronous fluorescence fingerprint represents 10 simple synchronous spectra placed in three-dimensional space with increment 20 nm, starting from Δλ = 10 nm. A constant wavelength synchronous map (CWM) was created automatically after scanning the synchronous spectra with FLWinLab software, Buckinghamshire, United Kingdom. For further analysis, synchronous spectra Δλ = 110 nm and synchronous profile were chosen and were arithmetically processed into the second derivative using FLWinLab software. The synchronous profile is represented graphically as the dependence of fluorescence on the excitation wavelength F = f(λ_Ex_). Further mathematical treatment of the second derivative of the synchronous profile spectrum provides two parameters that can improve the distinction between CTRL and POAG tear fluid. The synchronous fluorescence profile was generated using software developed by Martin Lešo, Ph.D. 

### 2.3. Statistical Analysis

Shapiro–Wilk test was used for testing of data normality. Unpaired two-sided t-test and Mann–Whitney U test were used to analyze differences between groups CTRL and PGOU according to data distribution. *p*-value < 0.05 was considered statistically significant.

## 3. Results

The paper presents results of different processing of 3D synchronous spectra of the same group of patients with POAG and control group (CTRL) as in the previously published paper [12]. The results were promising, so we have continued with mathematical processing of published 3D synchronous spectra data [12] to increase the resolution between the groups (POAG and CTRL). The results published in this paper differ from those previously published [12] in data processing.

At first, complex 3D synchronous spectra were processed in a simple curve F = f(λ_Ex_) as a synchronous excitation profile (SP) compared to previously published only as a single simple synchronous excitation spectrum Δλ = 110 nm. The synchronous profile (SP) was represented graphically as the dependence of maximum fluorescence intensity on the excitation wavelength F = f(λ_Ex_). Then the second derivative was applied to the synchronous excitation profile (2dSP). Finally, 2dSP was mathematically processed in an original way and displayed in a graph.

### 3.1. Total Synchronous Fluorescence

A series of constant wavelength simple synchronous spectra were arranged in sequence with increments of Δλ = 20 nm to form a fluorescent body. Total fluorescence is usually presented in the form of a contour map from a bird´s eye view. The contour map is complicated for visual comparison of samples, but very useful for the analysis of a tear fluid complex mixture.

To simplify further arithmetic processing, a front view of the fluorescent synchronous profile was used (Figure 1). The synchronous profile represents the maximum fluorescence intensity as a function of excitation wavelength. Another way to compare the composition of a complex mixture is to select one suitable synchronous spectrum from the 3D series. For a tear fluid, the best choice is a synchronous spectrum Δλ = 110 nm. The spectrum captures both fluorescent peaks, and it represents a complex mixture (Figure 2).

Both synchronous spectra Δλ = 110 nm and synchronous profile capture both fluorescence peaks, but neither of them allows reliable differentiation of glaucoma from control (Figure 3). Moreover, the synchronous spectrum Δλ = 110 nm does not capture the actual relationship between the fluorescent centers because it runs below the first and above the second fluorescent center. Also, the position of the excitation peak maxima does not correspond to the real excitation, and it is the same in the tear fluid of healthy subjects and in patients with glaucoma. In both cases (CTRL and POAG) the course of the spectra is almost identical, the differences are only in the fluorescence intensity of the second peak.

The fluorescence profile (SP) reflects the real relationship between the fluorescence centers, both quantitatively and qualitatively, but also the changes in the average spectra are not significant. The fluorescence profile shows only a difference in the wavelengths of the fluorescence peaks in the first fluorescence center.

### 3.2. “Data Mining” in Synchronous Spectra

To better distinguish the differences between the spectra (graphical representation of the function F = f(λ_Ex_) and to magnify the differences between them, mathematical derivative of the function was used.

#### 3.2.1. Derivative Synchronous Spectra

Derivative spectra are always more complex than zero-order spectra. The second-order derivative (2D) has been applied to the synchronous profile. The most characteristic feature of 2D is a negative band with a minimum at the same wavelength as the maximum of the fluorescence peak of the basis function (zero-order derivative) (Figure 4).

The second derivative was applied to a synchronous profile, not to the synchronous spectrum Δλ = 110 nm. The 2D synchronous spectrum increased differences between profiles of CTRL and POAG (Figure 5). The derivative spectrum consists of two concave curves, (denoted “+1” and “+2”) and two convex curves (“−1”; “−2”). The peaks “−1”, “+1”, and “−2” show the largest differences in the shape of the curve and/or the position of the minimum and maximum, respectively. The “−1” peak reflects the “redshift” of the POAG non-derivative synchronous profile wavelength (Figure 3).

The second fluorescence center of the non-derivative profile shows changes only in the increasing function, while the decreasing functions are almost identical. This is also reflected in the shape of the derivative spectrum. The “−2” peak is characterized by an altered shape, the “+2” peak of the derivative spectrum has the same shape, respectively; only slight changes were seen in both groups.

#### 3.2.2. Mathematical Processing of the Second Derivative Spectra of the Synchronous Profile

The differences in this graphic form are quite small, but inspiring for mathematical processing. The area of peaks of 2D spectra was mathematically processed. The differences in the first fluorescence center reflect the peaks “−1” and “+1”, in the second fluorescence center the peaks “−2” and “+2”. Different combinations of all peaks ratios of synchronous spectra have been tried. The best mathematical processing, which allowed a good graphical resolution of the control group from the glaucoma patients, is shown in Figure 6. The ratio of peaks of derivative spectra (“1”/“+1”)/(“−2”/“+2”) in the Figure 6 expresses the relationship between the two fluorescence peaks and characterizes the whole synchronous profile. In contrast, the ratio between the width of peaks “−1” and “+1” (“−1w”/“+1w”) represents the first fluorescence center where the differences between the CTRL and POAG groups were the greatest. F is factor calculated from data of the second derivative spectra. The procedure for calculating the value given as (“−1w”/“+1w”)*F is not explained in detail because it is being considered for patenting.

The procedure for calculating the value given as (“−1w”/“+1w”)*F is not explained in detail, it is being considered for patenting. 

#### 3.2.3. Statistical Analysis

According to the results of the statistical analysis (Table 2), in the case of the synchronous profile (SP), there is a significant differentiation of patients (POAG) and healthy individuals (CTRL) in both x and y axes. In contrast, the synchronous spectrum Δλ = 110 nm shows a significant result only in the y axis. The results confirm that the differentiation of patients and healthy subjects is improved by using the second derivative synchronous profile. 

## 4. Discussion

For a long time, the fluorescence analysis of body fluids has been a challenge for researchers. The ability to analyze multicomponent body fluids without pre-analytical sample processing is particularly useful for routine analyses. Endogenous fluorescence of biomolecules allows a variety of fluorescence techniques to be applied to the analysis of body fluids, including tear fluid.

The combination of synchronous fluorescence analysis and mathematical processing of synchronous spectra with zero order and second derivative, enhances the differences between samples with very similar compositions [11]. Derivation of the spectra reveals subtle hidden and visually imperceptible features of the spectral curve.

Our previously published results show a positive outcome when applying the second derivative to a selected synchronous spectrum Δλ = 110 nm which allows us to differentiate between healthy subjects (CTRL) and patients with glaucoma (POAG) [12]. We have decided to improve resolution between the CTRL and POAG groups. In this paper, we have mathematically analyzed synchronous fluorescence excitation spectra in a novel way. Three-dimensional synchronous fluorescence spectra (express total fluorescence of tear fluid) were used. The front view of a 3D fluorescent object was applied to simplify the comparison of different fluorescent complex mixtures [13]. With this new approach, we improved our ability to discriminate between POAG and CTRL group.

There are two massive fluorescence peaks with the different wavelength maximum on the synchronous profile (SP) of tear fluid. In contrast to the synchronous spectrum Δλ = 110 nm, the synchronous profile of tear fluid truly reflects the fluorescence intensity of the peaks and their excitation position. The differences in the spectra of the groups are slight. The mathematical processing of data by derivative analysis allows higher resolution of data. Proteins in tear fluid display two broad peaks. Peaks correspond to multiple overlapping bands arising from the aromatic rings of phenylalanine, tyrosine, and tryptophan residues of proteins present in tear fluid. With derivative analysis, it is possible to resolve tyrosine and phenylalanine fluorescence of the first peak of tear fluid with an excitation maximum λex = 220 nm (Figure 3) from tryptophan fluorescence of the second peak (λex = 278 nm) of tear fluid [10]. “Red shift” of the first fluorescence peak in POAG group (maximum λex = 223–226 nm) on the synchronous profile of tear fluid indicates changes in the composition of fluorophores, or the microenvironment of the given fluorophore. It may be caused by changes in metabolome of tear fluid of patients with glaucoma. The second derivative spectra increased the distinctions between the groups (Figure 5), and their mathematical processing excellently differentiated control group (CTRL) from patients with glaucoma (Figure 6). Results of statistical analysis (Table 2) confirm the graphical result.

## 5. Conclusions

Tear fluid is not a commonly used biological material in clinico–biochemical laboratories. Currently, the enormous diagnostic potential of such a small volume is analyzed only experimentally. This paper describes selective and computer-solved methods, namely, zero order and second derivative synchronous spectrofluorimetry. Three-dimensional synchronous fluorescence spectra of tear fluid were transformed into a synchronous fluorescence excitation profile (SP). Subsequently, a second derivative was applied to this profile, and the derivative spectrum was further mathematically processed, resulting in two parameters that can differentiate healthy subjects from glaucoma patients. The developed evaluation method was found to be selective and inexpensive for sensitive, rapid, and early qualitative determination of glaucoma in tear fluid in the laboratory at nano-levels. Combining different types of 3D-SFS data can offer interesting and useful tools for the diagnosis of glaucoma in tear fluid, and it can be used as input for machine and process automation.

## Figures and Tables

**Figure 1 sensors-22-05534-f001:**
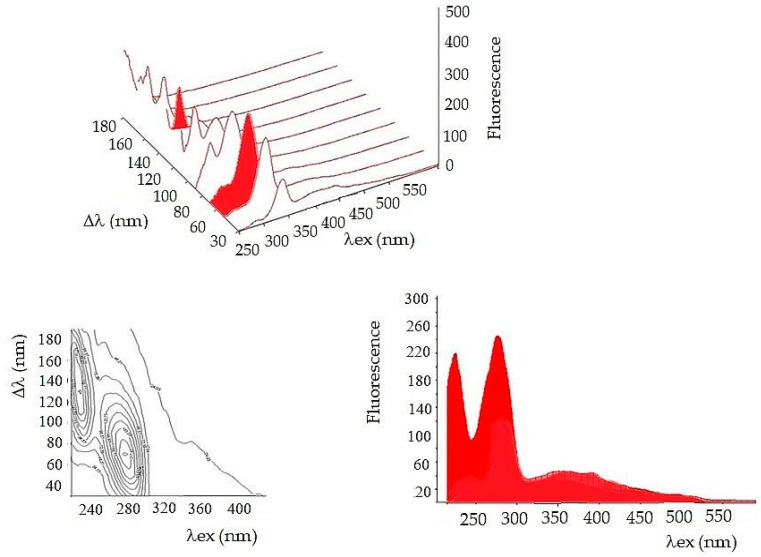
This is a figure of total tear fluorescence presented in different ways. **Upper**: Arrangement of spectra in a series in a 3D coordinate system. Axis x = excitation wavelength, y = Δλ, z = fluorescence. **Bottom right**: Synchronous profile (SP). **Bottom left**: Contour map—synchronous fluorescent map (CWM).

**Figure 2 sensors-22-05534-f002:**
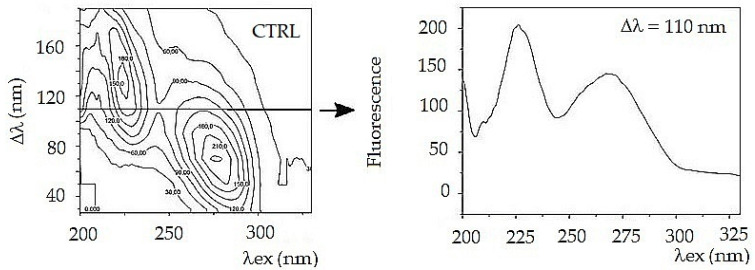
Constant wavelength synchronous map (CWM) tears of a healthy subject (**left**) with indication of a simple synchronous spectrum. It is shown on the **right** [12]. Authors have asked for permission to reprint Figure 5 from [12] and the request was approved.

**Figure 3 sensors-22-05534-f003:**
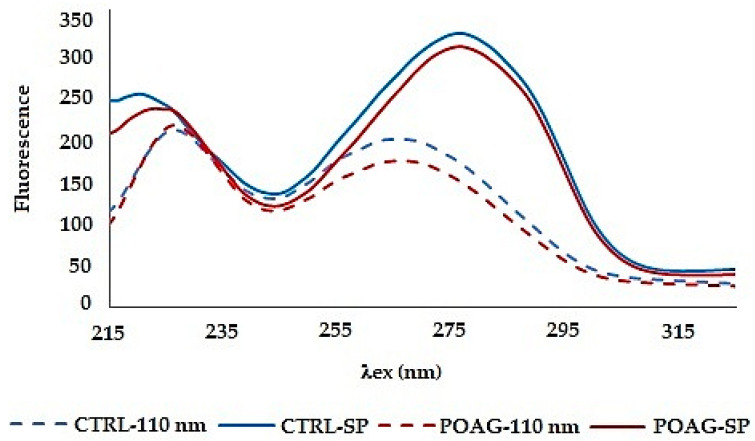
Comparison of SFS, Δλ = 110 nm and synchronous profile (SP) of selected groups. The average spectrum of tear sample´s group is shown: CTRL—control, POAG—glaucoma [12].

**Figure 4 sensors-22-05534-f004:**
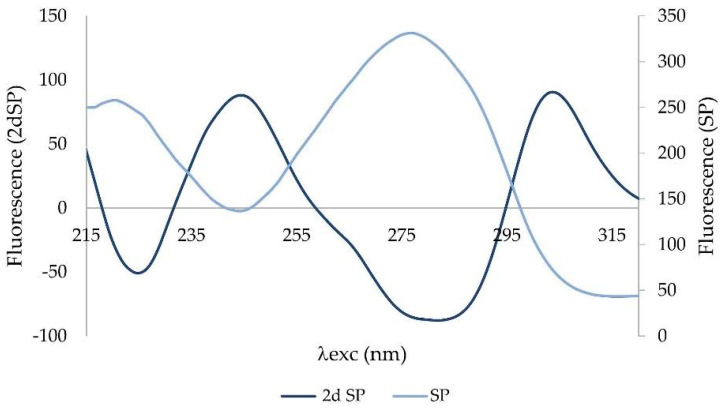
Synchronous profile (SP) of tear fluid and its second derivation (2dSP).

**Figure 5 sensors-22-05534-f005:**
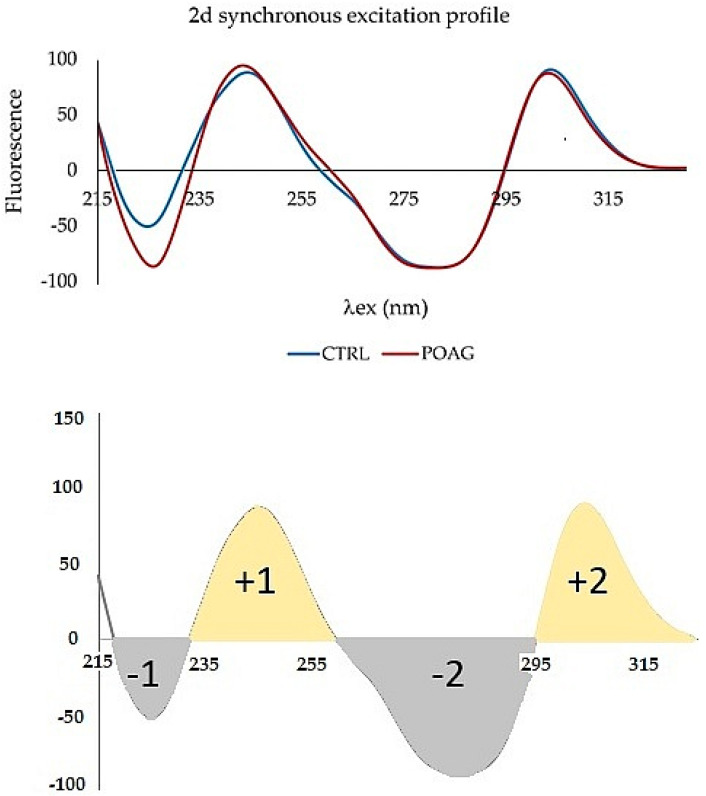
**Upper** graph: The second derivative of the synchronous spectrum of tear fluid of control group (CTRL) and patients with glaucoma (POAG). The spectra represent the average of the groups. **Bottom** graph: Zones of the second derivative of the synchronous spectrum. These were used to process the spectra of tear fluid. Mathematical processing was done to build plot in Figure 6.

**Figure 6 sensors-22-05534-f006:**
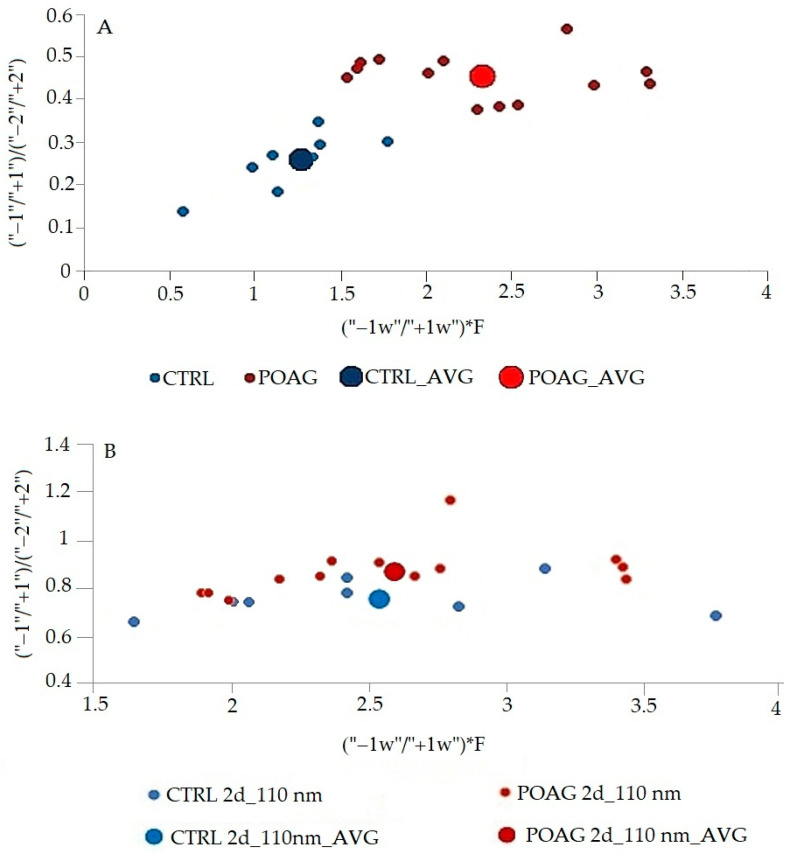
Comparison of the mathematical processing of two different ways of using 3D derivative fluorescence analysis to distinguish between glaucoma patients (POAG) and healthy subjects with normal intraocular pressure (CTRL). Graphical differentiation between tear fluid of glaucoma patients and healthy subjects with normal intraocular pressure. (**A**). Mathematically processed second derivative synchronous profile (SP) of glaucoma patients and healthy subjects tear fluid. CTRL_AVG. (**B**). Mathematically processed second derivative synchronous spectrum Δλ= 110 nm of glaucoma patients and healthy subjects tear fluid. The second derivative of both synchronous profile and synchronous spectrum Δλ = 110 nm were mathematically processed, and the result was plotted on a graph. The ratio (“1”/“+1”)/(“−2”/“+2”) expresses the relationship between the two fluorescence peaks and characterizes the whole synchronous profile. In contrast, the ratio (“−1w”/“+1w”) represents only the first fluorescence center where the differences between the CTRL and POAG groups were greatest. The large point represents the average (AVG) of the results in the group of POAG and CTRL. The same patients (POAG) and healthy subjects (CTRL) were used for the analyses of the synchronous profile and Δλ = 110 nm synchronous spectrum.

**Table 1 sensors-22-05534-t001:** Description of studied healthy subjects (CTRL) and patients with glaucoma POAG.

^1^ Average	^1^ CTRL	^1^ POAG
Patients	8	13
Males/Females	4/4	4/9
Age (years)	38 ± 11	56 ± 14
Intraocular pressure (torr)	15.7 ± 1.7	16.5 ± 4.7
Retinal nerve fiber layer (μm)	normal	99
Ganglion cell complex (μm)	normal	90
Medication	0	12
No Medication	8	1
Brimonide tartrate (BT)	-	2
Prostaglandins	-	-
Prostaglandin analogue (PA)	-	-
PA + Beta blockers	-	2
PA + Beta blockers + BT	-	1
PA + Beta blockers + Corticosteroids	-	1
Carboanhydride inhibitor (CI)	-	1
CI + Beta blockers	-	2
CI + Beta blockers + BT	-	1
CI + PA	-	2
CI + PA + Beta blockers	-	-

^1^ Selected data were taken with permission from paper [12].

**Table 2 sensors-22-05534-t002:** Results of statistical analysis of patients POAG and healthy subjects (CTRL).

	Axis	Data	Average ± STD	Test	*p*-Value	Result
SP	x: (“−1w”/“+1w”)*F	2d_SP_CTRL	1.23 ± 0.34	*t* test	0.000207	*** *p* < 0.001
		2d_SP_POAG	2.39 ± 0.62			
	y: (“−1”/“+1”)/(“−2”/“+2”)	2d_SP_CTRL	0.25 ± 0.06	*t* test	<0.00001	***** *p* < 0.00001
		2d_SP_POAG	0.44 ± 0.05			
Δλ = 110	x: (“−1w”/“+1w”)*F	2d_110_CTRL	2.55 ± 0.61	*t* test	0.84	*p >* 0.05
		2d_110_POAG	2.6 ± 0.54			
	y: (“−1”/“+1”)/(“−2”/“+2”)	2d_110_CTRL	0.79 ± 0.07	Mann–Whitney U test	0.0083	** *p* < 0.01
		2d_110_POAG	0.93 ± 0.11			

STD Standard deviation, Significant, ***** *p* < 0.00001, *** *p* < 0.001, ** *p* < 0.01 are significant, *p >* 0.05 is insignificant.

## Data Availability

Not applicable.

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
