# Peer review of "Derivative Three-Dimensional Synchronous Fluorescence Analysis of Tear Fluid and Their Processing for the Diagnosis of Glaucoma"

_sensors, 2022, doi:10.3390/s22155534_

Round 1
Reviewer 1 Report
Dubayová et al. make an interesting report of using tear film analysis to distinguish between POAG and control patients. Their follow-up study results are interesting, timely, and believable. I do have a relatively big objection to the manuscript overall. Nowhere in the Results nor the front end of the Discussion do the authors say something like: "This technique improves our ability to discriminate between POAG patients and controls." I HIGHLY recommend they do so. The authors do so in the Conclusion, but it would help the flow if it were also stated earlier.
Additionally, here are a few more minor comments.
1. MINOR: Line 13 (and other places) - Why "CTR" for normal? I suggest at LEAST changing to CTRL (to match POAG).
2. MINOR: Line 25 - Please change to "...the whole body..."
3. MINOR: Lines 36-37 - Please consider changing to "...more diverse compared to..." (if I'm understanding correctly).
4. MINOR: Line 38 - Please change to: "...for specificity and reliability of diagnostic tests and treatment targets."
5. MINOR: Line 49 - Please change to: "...procedures but is sensitive to changes caused by pathological conditions is crucial."
6. MINOR: Line 69 - Consider changing to: "In this article, synchronous ..." (i.e., add comma)
7. MINOR: Line 71 - Consider adding comma here: "...procedures, ..."
8. MINOR: Line 74 - Please make this [F = f(lex)] consistent with Line 115.
9. MINOR: Line 99 - Consider adding comma here: "... fluorescent peaks, and it represents..."
10. MAJOR: Line 144 - Please change "The ratio (“1/+1”)/(-2/+2) expresses ..." to "The ratio (“1"/"+1”)/("-2"/"+2") expresses ..." (or something that is more consistent with the earlier text).
11. MINOR: Line 147 - This comment seems out of place and/or needs a reference or more details concerning Figure 6.
12. MINOR/STYLE SUGGESTION: Lines 153-157 - I recommend removed line breaks and adding commas between Upper..., Bottom Left..., and Bottom Right...
13. MINOR: Line 159: Please change Figure 2 legend to "...constant wavelength synchronous map (CWM)..." If a reader goes directly to figures, they will no know what 'CWM' means.
14. MINOR: Line 172 - Please consider a change to "...patients and healthy subjects with ..." (i.e., add 's' to end of subject)
15. MINOR: Line 166 - Please consider changing Figure legend to "...Synchronous profile (SP) of tear fluid..."
16. MAJOR: Line 194 - I do not think this adds anything: "Front view was introduced to simplify comparison of different fluorescent complex mixture." Consider removing.
17. MAJOR: Line 210 - This statement is awkward and hard to follow the meaning: "Tear fluid is not common body fluid which is studied in clinico–biochemical laboratory." Consider changing to something like, "Tear fluid is not commonly studied studied in clinicobiochemical laboratories."
18. MAJOR: Line 214 - Please consider changing to: "...profile (SP). A second derivative was then..." (i.e., separate into two sentences)
Author Response
Response to Reviewer 1 Comments
Dear reviewer ,
we appreciate the time and effort that you have dedicated to providing your valuable feedback on our manuscript. We are grateful to you for your insightful comments on our manuscript that will increase the paper quality. Here is a point-by-point response to the reviewers’ comments and concerns.
With best regards,
authors
Assoc. prof. Katarína Dubayová, PhD., RNDr., Kristína Krajčíková, PhD., prof. Mária Mareková, CSc and Assoc prof. Vladimíra Tomečková, PhD. (Corresponding author)
Reviewer 1:
Point 1: Dubayová et al. make an interesting report of using tear film analysis to distinguish between POAG and control patients. Their follow-up study results are interesting, timely, and believable. I do have a relatively big objection to the manuscript overall. Nowhere in the Results nor the front end of the Discussion do the authors say something like: "This technique improves our ability to discriminate between POAG patients and controls." I HIGHLY recommend they do so. The authors do so in the Conclusion, but it would help the flow if it were also stated earlier.
Response 1 (point1): We added a sentence to Discussion: “With this new approach, we improved our ability to discriminate between POAG and CTRL group.”
Point 2: Additionally, here are a few more minor comments.
- MINOR: Line 13 (and other places) - Why "CTR" for normal? I suggest at LEAST changing to CTRL (to match POAG).
Response 1 (point 2): Your suggestion was accepted CTR was rewrote to CTRL; CTRL was used in graphs and in text for healthy subjects.
Point 3: MINOR 2: Line 25 - Please change to "...the whole body..."
Response 1 (point 3): The sentence was changed as suggested: “Composition of tear fluid has diagnostic potential, as it provides information about health and pathologic state of the eyes, but also function of the whole body.”
Point 4: MINOR 3: Lines 36-37 - Please consider changing to "...more diverse compared to..." (if I'm understanding correctly).
Response 1 (point 4): The sentence was changed as suggested: “The aqueous layer is the most diverse tear film layer compared to the lipid and mucous layer.”
Point 5: MINOR 4: Line 38 - Please change to: "...for specificity and reliability of diagnostic tests and treatment targets."
Response 1 (point 5): The sentence was changed as suggested: “The source of biomarkers is crucial for specificity and reliability of diagnostic tests and treatment targets.”
Point 6: MINOR 5: Line 49 - Please change to: "...procedures but is sensitive to changes caused by pathological conditions is crucial."
Response 1 (point 6): The sentence was changed as suggested: The use of an appropriate analytical method that characterizes its composition without separation procedures but is sensitive to changes caused by pathological conditions is crucial.
Point 7: MINOR 6: Line 69 - Consider changing to: "In this article, synchronous ..." (i.e., add comma)
Response 1 (point 7): In the above-mentioned sentence, a comma was added.
Point 8: MINOR 7: Line 71 - Consider adding comma here: "...procedures, ..."
Response 1 (point 8): In the above-mentioned sentence, a comma was added.
Point 9: MINOR 8: Line 74 - P lease make this [F = f(lex)] consistent with Line 115.
Response 1 (point 9): The equation was unified as F = f(λEx).
Point 10: MINOR 9: Line 99 - Consider adding comma here: "... fluorescent peaks, and it represents..."
Response 1 (point10): In the above-mentioned sentence, a comma was added.
Point 11: MAJOR 10: Line 144 - Please change "The ratio (“1/+1”)/(-2/+2) expresses ..." to "The ratio (“1"/"+1”)/("-2"/"+2") expresses ..." (or something that is more consistent with the earlier text).
Response 1 (point11): The sentence was changed as suggested: (“1“/“+1”)/(“-2“/“+2“)
Point 12: MINOR 11: Line 147 - This comment seems out of place and/or needs a reference or more details concerning Figure 6.
Response 1 (point 12):
We added more details in caption of Figure 6 as you suggested: Figure 6. Graphical differentiation between tear fluid of glaucoma patients and healthy subjects with normal intraocular pressure. The 2nd derivative spectra of tear fluid were mathematically processed, and the result was plotted on a graph. The ratio (“1“/“+1”)/(“-2“/“+2“) expresses the relationship between the two fluorescence peaks and characterizes the whole synchronous profile. In contrast, the ratio (´-1w/+1w´) represents only the first fluorescence center where the differences between the CTRL and POAG groups were greatest.
We added more details in the text:
The differences in this graphic form are quite small, but inspiring for mathematical processing. Area of peaks of 2d spectra were mathematically processed. The differences in the first fluorescence center reflect the peaks “-1” and “+1”, in the second fluorescence center the peaks “-2” and “+2”. Different combinations of all peaks ratios of synchronous spectra have been tried. The best mathematical processing, which allowed a good graphical resolution of the control group from the glaucoma patients, is shown in Figure 6. The ratio (“1“/“+1”)/(“-2“/“+2“) in the Figure 6 expresses the relationship between the two fluorescence peaks and characterizes the whole synchronous profile. In contrast, the ratio (´-1w/+1w´) represents only the first fluorescence center where the differences between the CTRL and POAG groups were the greatest. Many combinations of peak-to-peak The resolution in this mathematical processing is excellent (Figure 6).
Point 13: MINOR 12/STYLE SUGGESTION: Lines 153-157 - I recommend removed line breaks and adding commas between Upper..., Bottom Left..., and Bottom Right...
Response 1 (point 13): The legend of Figure 1 was updated as suggested: Upper: Arrangement of spectra in a series in a 3D coordinate system. Axis x = excitation wavelength, y = Δλ, z = fluorescence., Bottom right: Synchronous profile (SP)., Bottom left: Contour map – synchronous fluorescent map (CWM).
Point 14: MINOR 13: Line 159: Please change Figure 2 legend to "...constant wavelength synchronous map (CWM)..." If a reader goes directly to figures, they will no know what 'CWM' means.
Response 1 (point 14): Figure 2 legend was updated according to the suggestion: A constant wavelength synchronous map (CWM) was created automatically after scanning the synchronous spectra with FLWinLab software.
Point 15: MINOR 14: Line 172 - Please consider a change to "...patients and healthy subjects with ..." (i.e., add 's' to end of subject)
Response 1 (point 15): The caption (Figure 6) was changed as suggested: Graphical differentiation between tear fluid of glaucoma patients and healthy subjects with normal intraocular pressure.
Point 16: MINOR 15: Line 166 - Please consider changing Figure legend to "...Synchronous profile (SP) of tear fluid..."
Response 1 (point 16): Figure 4 legend was changed as suggested: “Synchronous profile (SP) of tear fluid and its second derivation (2d SP).”
Point 17: MAJOR 16: Line 194 - I do not think this adds anything: "Front view was introduced to simplify comparison of different fluorescent complex mixture." Consider removing.
Response 1 (point 17): Sentence "Front view was introduced to simplify comparison of different fluorescent complex mixture" was removed according to your suggestion.
We slightly modified sentence: Three-dimensional synchronous fluorescence spectra (express total fluorescence of tear fluid) were used, and the front view was applied on a 3D fluorescent object to simplify comparison of different fluorescent complex mixture [13]. With this new approach, we improved our ability to discriminate between POAG and CTRL group.
Point 18: MAJOR 17: Line 210 - This statement is awkward and hard to follow the meaning: "Tear fluid is not common body fluid which is studied in clinico–biochemical laboratory." Consider changing to something like, "Tear fluid is not commonly studied studied in clinicobiochemical laboratories."
Response 1 (point 18): To clarify meaning, the sentence was changed: Tear fluid is not a commonly used biological material in clinico–biochemical laboratory.
Point 19: MAJOR 18: Line 214 - Please consider changing to: "...profile (SP). A second derivative was then..." (i.e., separate into two sentences)
Response 1 (19): The sentence was changed as suggested: 3 D synchronous fluorescence spectra of tear fluid were transformed into a synchronous fluorescence excitation profile (SP). Subsequently, a second derivative was applied to this profile, and the derivative spectrum was further mathematically processed resulting in two parameters that can differentiate healthy subjects from glaucoma patients.

Reviewer 2 Report
An interesting topic and a possible future application of a new strategy of a newly introduced method to diagnose glaucoma based on tear film fluid sample analysis.
The weak points of the present article are significant and reduce the validity of the conclusions therefore must be improved.
1. The numbers of glaucoma and control study subjects are very low. Need to perform a statistical power analysis to be known the necessary sample size enough to draw any conclusions.
2. The presentation of the data of study subjects is insufficient. Need to know their demographics (age, sex) and that the glaucoma and control groups are similar and statistically comparable.
3. As the general health status might cause changes in tear fluid composition therefore there is a need to prove that the general health conditions of the glaucoma group are similar to the control group.
4.a. The description of the glaucoma group needs to be improved, especially mentioning any possible artificial effects which might alter the tear film composition, like: glaucoma therapy (use of eye drops or untreated), any invasive eye intervention(s) before tear sample collection in the study, like e.g. intraocular pressure measurement or operation.
4.b. The description of the glaucoma group needs to be improved, if there is any other eye disease or not which might alter the tear film fluid.
4.c. The description of the glaucoma group needs to be improved, to let to know the stage of their glaucoma disease of the study participants (early glaucoma or not?, visual field defects, nerve fiber layer defects, etc) and the basis/criteria of their glaucoma diagnosis.
Author Response
Response to Reviewer 2 Comments
Dear reviewer ,
we appreciate the time and effort that you have dedicated to providing your valuable feedback on our manuscript. We are grateful to you for your insightful comments on our manuscript that will increase the paper quality. Here is a point-by-point response to the reviewers’ comments and concerns.
Reviewer 2:
An interesting topic and a possible future application of a new strategy of a newly introduced method to diagnose glaucoma based on tear film fluid sample analysis. The weak points of the present article are significant and reduce the validity of the conclusions therefore must be improved.
Point 1: The numbers of glaucoma and control study subjects are very low. Need to perform a statistical power analysis to be known the necessary sample size enough to draw any conclusions.
Response 2 (point1): This project is a pilot project, so the number of patients and controls is small. The obtaining enough of tear fluid from patients with glaucoma was many times difficult or impossible. Many patients had lowered production of tear fluid. These patients were avoided from the study. This is an initial study of expanding the diagnostic possibilities of glaucoma based on tear analysis. 3D synchronous fluorescence spectroscopy was applied to image the complex native composition of tear fluorophores. There were 2 distinct fluorescence centers on the constant wavelength synchronous map, which corresponded to the fluorescence of aromatic amino acids based on their fluorescence characteristics. In the 1st stage of study (Spectroscopy letters publication), a synchronous spectrum of Dl =110 nm was chosen to simplify the 3D recording. This record captures both centers, but not in their peak. In the 2nd stage (submitted publication in Sensors) there was an effort to simplify the 3D recording so that it corresponded to the real fluorescence of both centers, therefore the profile (frontal view) was chosen for the 3D spectra. In the 1st stage of the 2nd derivation of the synchronous spectrum brought encouraging results in the ability to distinguish a group of healthy, non-glaucoma patients (CTRL = control group) from patients with POAG. Therefore, the 2nd derivation spectroscopy was also applied to the fluorescence profile in the present work. Due to the initial study on a small sample of subjects, we considered the statistical analysis irrelevant. The aim of the work was to point out the possibility of finding a mathematical algorithm for the processing of derived spectra, which would highlight the differences between groups, and it allows the given groups to be distinguished.
Statistical analysis was added in Material and Methods in manuscript.
2.1 Statistical analysis
Shapiro-Wilk test was used for testing of data normality. Mann-Whitney U test and Unpaired one-sided t-test were used to analyze differences between groups CTRL and PGOU for synchronous spectrum Δλ = 110 nm and derivative synchronous spectrum, respectively, p-Value < 0.05 was considered statistically significant.
Statistical analysis was added:
Table 2. Results of statistical analysis of patients POAG and healthy subjects (CTRL)
|
|
Axis |
Data |
Average |
STD |
Test |
p-value |
Result |
|
SP |
x: ("-1w"/"+1w")*F |
2d_SP_CTRL |
1.23 |
0.34 |
t test |
0.000207 |
significant |
|
2d_SP_POAG |
2.39 |
0.62 |
|||||
|
y: ("-1"/"+1")/("-2"/"+2") |
2d_SP_CTRL |
0.25 |
0.06 |
t test |
< 0.00001 |
significant |
|
|
2d_SP_POAG |
0.44 |
0.05 |
|||||
|
Δλ = 110 |
x: ("-1w"/"+1w")*F |
2d_110_CTRL |
2.55 |
0.61 |
t test |
0.84 |
insignificant |
|
2d_110_POAG |
2.6 |
0.54 |
|||||
|
y: ("-1"/"+1")/("-2"/"+2") |
2d_110_CTRL |
0.79 |
0.07 |
Mann-Whitney U test |
0.0083 |
significant |
|
|
2d_110_POAG |
0.93 |
0.11 |
|||||
|
|
|
|
|
|
|
|
|
Point 2: The presentation of the data of study subjects is insufficient. Need to know their demographics (age, sex) and that the glaucoma and control groups are similar and statistically comparable.
Response 2 (point 2): Demographics (age, sex) was described in published paper [12]. We accepted your suggestion and we added in Material and Methods Table 1 with selected data of study.
Table 1. Description of studied healthy subjects (CTRL) and patients with glaucoma POAG.
|
Average |
CTRL |
POAG |
|
Patients |
8 |
13 |
|
Males/Females |
4/4 |
4/9 |
|
Age (years) |
38 ± 11 |
56 ± 14 |
|
Intraocular pressure (torr) |
15.7 ± 1.7 |
16.5 ± 4.7 |
|
Retinal nerve fiber layer (μm) |
normal |
99 |
|
Ganglion cell complex (μm) |
normal |
90 |
|
Medication |
0 |
12 |
|
No Medication |
10 |
1 |
|
Brimonide tartrate (BT) |
- |
2 |
|
Prostaglandins |
- |
- |
|
Prostaglandin analogue (PA) |
- |
- |
|
PA + Beta blockers |
- |
5 |
|
PA + Beta blockers + BT |
- |
1 |
|
PA + Beta blockers + Corticosteroids |
- |
1 |
|
Carboanhydride inhibitor (CI) |
- |
1 |
|
CI + Beta blockers |
- |
2 |
|
CI + Beta blockers + BT |
- |
1 |
|
CI + PA |
- |
2 |
|
CI + PA + Beta blockers |
- |
- |
1 Selected data were taken with permission from paper [12]
Point 3: As the general health status might cause changes in tear fluid composition therefore there is a need to prove that the general health conditions of the glaucoma group are similar to the control group.
Response 2 (point 3): Healthy subjects were without medication, without ocular disease, without loss of tear production, without glaucoma, dry eye syndrome (DES), systemic diseases, such as diabetes mellitus, major depressive disorder or even cancer. Patients were selected with lower age than 40 years old. The diagnosis of glaucoma and related diseases in patients was based on the guidelines from the European Glaucoma Society (European Glaucoma Society Terminology and Guidelines for Glaucoma, 4th). The obtaining tear fluid from patients with glaucoma was many times difficult or impossible. Many patients had lowered production of tear fluid. These patients were avoided from the study. Patients with loss of tear production, without glaucoma, dry eye syndrome (DES), systemic diseases, such as diabetes mellitus, major depressive disorder or even cancer, with another type of glaucoma were also excluded from the study.
Point 4.a. The description of the glaucoma group needs to be improved, especially mentioning any possible artificial effects which might alter the tear film composition, like: glaucoma therapy (use of eye drops or untreated), any invasive eye intervention(s) before tear sample collection in the study, like e.g. intraocular pressure measurement or operation.
Response 2 (point 4.a): Patients were selected without operation. They were treated with their glaucoma therapy (Table 1). They used usually eye drops or/and also oral treatment with drugs.
Point 4.b. The description of the glaucoma group needs to be improved, if there is any other eye disease or not which might alter the tear film fluid.
Response 2 (point 4.b): Patients with another eye disease were excluded from the study.
Point 4.c. The description of the glaucoma group needs to be improved, to let to know the stage of their glaucoma disease of the study participants (early glaucoma or not? Visual field defects, nerve fiber layer defects, etc) and the basis/criteria of their glaucoma diagnosis.
Response 2 (4.c): Patients with primary open angle glaucoma were selected for this study. The description of glaucoma is in Table 1 which was added in Material and Methods.
We collected tear fluid also from patients with different stage of glaucoma (Table X): glaucoma with open angle (POAG), suspected glaucoma (SUSP GLAU) and from the patients with ocular hypertension (OHT) in published paper [12]. Comparison of healthy subject with patients with glaucoma was selected in the current paper.
Table X Patient`s description and characteristics in published paper [12]
|
|
CTRL |
POAG |
OHT |
SUSP GLAU |
|
Patients |
8 |
13 |
12 |
7 |
|
Males/Females |
4/4 |
4/9 |
4/8 |
3/4 |
|
Mean age (years) |
38 |
56 |
33 |
42 |
|
Mean intraocular pressure (torr) |
17 |
17 |
19 |
15 |
|
Average retinal nerve fiber layer (μm) |
normal |
99 |
108 |
105 |
|
Average ganglion cell complex (μm) |
normal |
90 |
91 |
97 |
|
Medication |
0 |
12 |
9 |
0 |
|
Without medication |
8 |
1 |
3 |
7 |
|
Brimonide tartrate (BT) |
- |
2 |
- |
- |
|
Prostaglandins |
- |
- |
3 |
- |
|
Prostaglandin analogue (PA) |
- |
- |
2 |
- |
|
PA + Beta blockers |
- |
5 |
1 |
- |
|
PA + Beta blockers + BT |
- |
1 |
- |
- |
|
PA + Beta blockers + Corticosteroids |
- |
1 |
- |
- |
|
Carboanhydride inhibitor (CI) |
- |
1 |
1 |
- |
|
CI + Beta blockers |
- |
2 |
- |
- |
|
CI + Beta blockers + BT |
- |
1 |
- |
- |
|
CI + PA |
- |
2 |
1 |
- |
|
CI + PA + Beta blockers |
- |
- |
1 |
- |
We would like to thank the reviewer for his stimulating comments.
With best regards,
authors
Assoc. prof. Katarína Dubayová, PhD., RNDr., Kristína Krajčíková, PhD., prof. Mária Mareková, CSc and Assoc prof. Vladimíra Tomečková, PhD. (Corresponding author)

Reviewer 3 Report
In their original research article, Katarína Dubayová and colleagues explore the use of three-dimensional synchronous fluorescence spectroscopy as a tool for the ready diagnosis of glaucoma. The Authors introduce a new approach that seems to be effective in the distinction between healthy subjects and patients with primary open-angle glaucoma. Prior to publishing however, the Authors should address several concerns in order to improve the readability, impact and consistency of their work.
Broad comments
11) Figure 2 is not new and is exactly the same as Figure 5 in reference 12. The Authors should make it clear that these data have been presented elsewhere. Moreover, it is not mentioned in the legend of Figure 2 if the Authors have asked for permission to reprint it.
22) Authors state that the second order derivative (2d) of the synchronous profile increases differences between profiles of CTR and POAG, which is shown in Figure 5. According to their previous work (reference 12), the 2d of the synchronous spectrum Δλ = 110 nm (Figure 7 in that paper) also increases the differences between CTR and POAG. Considering this, it is my understanding that, as the manuscript stands, the Authors fail to show how the synchronous profile is advantageous over the synchronous spectrum to make the distinction between CTR and POAG. I would suggest the authors to apply a similar mathematical process to the one they used to generate Figure 6 to the 2d of the synchronous spectrum Δλ = 110 nm to unequivocally show that drawing the synchronous profile is advantageous over the synchronous spectrum.
33) Supplementary Figures are not mentioned in the manuscript, only under Supplementary Materials. It is my understanding that Supplementary Figures 7 and 9 should be part of the main text. Supplementary Figure 7 can be included in Figure 5, which will help the reader to easily understand the mathematical processing done to build plot in the current Figure 6. Supplementary Figure 9, on the other hand, can be included in Figure 6. In Supplementary Figure 8 and 9 the Authors present data concerning patients that mentioned there for the first time. The Authors should state in the Material and Methods section that data from patients with those pathologies will also be analysed using the same methodology and they should present the synchronous profile of those samples as well.
Data in Supplementary Figure 9 is presented as average. Is the size of the data point proportional to the mean ± standard deviation?
Regarding data in Supplementary Figure 8, that is presented in Supplementary Figure 9 as average, what if the authors would perform a linear regression to describe to behaviour of each set of samples? Considering the methodology proposed by the Authors, would this be a valid approach?
44) Lines 199 – 201: “Fluorescence of the… tryptophan fluorescence [10].” Please rephrase this sentence considering that tyrosine is also excited at 278 nm (with emission maximum close to 310 nm in opposition to 340 nm as in the case of tryptophan), although its molar extinction coefficient is lower than that of tryptophan at this wavelength.
55) The manuscript can benefit from a more detailed description of the fluorescence analysis, regarding equipment, slits, cuvettes used and other important details.
66) Although the differences between samples from healthy subjects and from patients with primary open-angle glaucoma seems to be clear, statistical analysis is missing.
Specific comments
11) Lines 86 – 88: These are the journal guidelines and need to be deleted.
22) Line 99: “Both,” – remove comma
33) Lines 126-127: “The peaks “-1”; “+1” and “-2” show the largest differences in the shape…” – Do the Authors mean “-2” or “+2”? My question is in the sense that by visual inspection it seem that the peak “+2” shows a larger difference than the peak “-2”.
44) Line 144: “The ratio (“1/+1”)/(-2/+2) expresses” – Add “-“ before the 1.
55) Legend of Figure 6 must be complemented with the information concerning the mathematical process behind the data presented so that the reader does not need to check the manuscript to understand this figure.
16) Some proofreading and English correction is necessary.
Author Response
Response to Reviewer 3 Comments
Dear reviewer ,
we appreciate the time and effort that you have dedicated to providing your valuable feedback on our manuscript. We are grateful to you for your insightful comments on our manuscript that will increase the paper quality. Here is a point-by-point response to the reviewers’ comments and concerns.
Reviewer 3
In their original research article, Katarína Dubayová and colleagues explore the use of three-dimensional synchronous fluorescence spectroscopy as a tool for the ready diagnosis of glaucoma. The Authors introduce a new approach that seems to be effective in the distinction between healthy subjects and patients with primary open-angle glaucoma. Prior to publishing however, the Authors should address several concerns in order to improve the readability, impact and consistency of their work.
Point 1) Broad comments. Figure 2 is not new and is exactly the same as Figure 5 in reference 12. The Authors should make it clear that these data have been presented elsewhere. Moreover, it is not mentioned in the legend of Figure 2 if the Authors have asked for permission to reprint it.
Response 3 (point 1): We added in caption of Figure 2 information about permission to reprint the Figure 5 from reference 12.
Figure 2. Constant wavelength synchronous map (CWM) tears of a healthy subject (left) with indication of a simple synchronous spectrum. It is shown on the right [12]. Authors have asked for permission to reprint Figure 5 from [12] and the request was approved.
Point 2) Authors state that the second order derivative (2d) of the synchronous profile increases differences between profiles of CTR and POAG, which is shown in Figure 5. According to their previous work (reference 12), the 2d of the synchronous spectrum Δλ = 110 nm (Figure 7 in that paper) also increases the differences between CTR and POAG. Considering this, it is my understanding that, as the manuscript stands, the Authors fail to show how the synchronous profile is advantageous over the synchronous spectrum to make the distinction between CTR and POAG. I would suggest the authors to apply a similar mathematical process to the one they used to generate Figure 6 to the 2d of the synchronous spectrum Δλ = 110 nm to unequivocally show that drawing the synchronous profile is advantageous over the synchronous spectrum.
Response 3 (point 2): A new Figure 6 was created as per your suggestion and a plot of POAG patients and healthy subjects (CTRL) differentiated based on a single synchronous spectrum was added. the benefit of the synchronous profile was also confirmed statistically (Table 2)
Figure 6. Comparison of the mathematical processing of two different ways of using 3D derivative fluorescence analysis to distinguish between glaucoma patients (POAG) and healthy subjects with normal intraocular pressure (CTRL). Graphical differentiation between tear fluid of glaucoma patients and healthy subjects with normal intraocular pressure.
- Mathematically processed the 2nd derivative synchronous profile (SP) of glaucoma patients and healthy subjects tear fluid. CTRL_AVG. B. Mathematically processed the 2nd derivative synchronous spectrum Δλ= 110 nm of glaucoma patients and healthy subjects tear fluid.
The 2nd derivative of both synchronous profile and synchronous spectrum Δλ= 110 nm were mathematically processed, and the result was plotted on a graph. The ratio (“1“/“+1”)/(“-2“/“+2“) expresses the relationship between the two fluorescence peaks and characterizes the whole synchronous profile. In contrast, the ratio (“-1w”/”+1w”) represents only the first fluorescence center where the differences between the CTRL and POAG groups were greatest.
The large point represents the average (AVG) of the results in the group of POAG and CTRL
The same patients (POAG) and healthy subjects (CTRL) were used for the analyses of the synchronous profile and Δλ= 110 nm synchronous spectrum.
Point 3) Supplementary Figures are not mentioned in the manuscript, only under Supplementary Materials. It is my understanding that Supplementary Figures 7 and 9 should be part of the main text. Supplementary Figure 7 can be included in Figure 5, which will help the reader to easily understand the mathematical processing done to build plot in the current Figure 6. Supplementary Figure 9, on the other hand, can be included in Figure 6. In Supplementary Figure 8 and 9 the Authors present data concerning patients that mentioned there for the first time. The Authors should state in the Material and Methods section that data from patients with those pathologies will also be analysed using the same methodology and they should present the synchronous profile of those samples as well. Data in Supplementary Figure 9 is presented as average. Is the size of the data point proportional to the mean ± standard deviation? Regarding data in Supplementary Figure 8, that is presented in Supplementary Figure 9 as average, what if the authors would perform a linear regression to describe to behaviour of each set of samples? Considering the methodology proposed by the Authors, would this be a valid approach?
Response 3 (point 3): Supplementary Figures 7 and 9 was included in the main text according to your suggestions. Supplementary Figure 7 was included in Figure 5. Supplementary Figure 9 was included in Figure 6.
Figure 5
Figure 6
Point 4) Lines 199 – 201: “Fluorescence of the… tryptophan fluorescence [10].” Please rephrase this sentence considering that tyrosine is also excited at 278 nm (with emission maximum close to 310 nm in opposition to 340 nm as in the case of tryptophan), although its molar extinction coefficient is lower than that of tryptophan at this wavelength.
Response 3 (point 4): We rephrased this sentence in manuscript: “The mathematical processing of data by derivative analysis allows higher resolution of data. Proteins in tear fluid display two broad peaks. Peaks correspond to multiple overlapping bands arising from the aromatic rings of phenylalanine, tyrosine, and tryptophan residues of proteins present in tear fluid. With derivative analysis, it is possible to resolve tyrosine and phenylalanine fluorescence of the first peak of tear fluid with excitation maximum lex = 220 nm (Figure 3) from tryptophan fluorescence of the second peak (lex = 278 nm) of tear fluid [10].!
Point 5) The manuscript can benefit from a more detailed description of the fluorescence analysis, regarding equipment, slits, cuvettes used and other important details.
Response 3 (point 5): We added more detailed description of the fluorescence analysis, equipment, slits, cuvettes according to your recommendation in Material and Methods.
“Tear fluid was stored at -80 °C until fluorescence analysis was carried out in quartz dark microcuvette on Perkin-Elmer Luminescence Spectrophotometer LS 55. The setting of the intrument´s excitation and emission slits was 5, 5 nm. The synchronous fluorescence spectra were analyzed in range from 200-600 nm with scan rate 1200 nm/min. The measured data were processed by WinLab Software. Synchronous fluorescence fingerprint represents 10 simple synchronous spectra (Dl = 10 nm) placed in 3-dimensional space with increment 20 nm. The synchronous spectrum is represented graphically as the dependence of fluorescence on the excitation wavelength F = f(lEx). A constant wavelength synchronous map (CWM) was created automatically after scanning the synchronous spectra with FLWinLab software. For further analysis, synchronous spectra Dl = 110 nm and synchronous profile were chosen and were arithmetically processed into the second derivative also using FLWinLab software. Further mathematical treatment of the second derivative of the synchronous profile spectrum provides two parameters that can improve the distinction between CTRL and POAG tear fluid. The synchronous fluorescence profile was generated using software developed by Martin Lešo, PhD.”
Point 6) Although the differences between samples from healthy subjects and from patients with primary open-angle glaucoma seems to be clear, statistical analysis is missing.
Response 3 (point 6): This project is a pilot project, so the number of patients and controls is small. This is an initial study of expanding the diagnostic possibilities of glaucoma based on tear analysis. 3D synchronous fluorescence spectroscopy was applied to image the complex native composition of tear fluorophores. There were 2 distinct fluorescence centers on the cw contour map, which corresponded to the fluorescence of aromatic amino acids based on their fluorescence characteristics. In the 1st stage of study (Spectroscopy letters publication), a synchronous spectrum of 110 nm was chosen to simplify the 3D recording. This record captures both centers, but not in their peak. In the 2nd stage (submitted publication in Sensors) there was an effort to simplify the 3D recording so that it corresponded to the real fluorescence of both centers, therefore the profile (frontal view) was chosen for the 3D spectra. In the 1st stage of the 2nd derivation of the synchronous spectrum brought encouraging results in the ability to distinguish a group of healthy, non-glaucoma patients (CTR = control group) from patients with POAG. Therefore, the 2nd derivation spectroscopy was also applied to the fluorescence profile in the present work. Due to the initial study on a small sample of subjects, we considered the statistical analysis irrelevant. The aim of the work was to point out the possibility of finding a mathematical algorithm for the processing of derived spectra, which would highlight the differences between grous and it allows the given groups to be distinguished.
The work was presented as "Communication" for the stated reasons. Research in this area continues and after obtaining a sufficient number of patients and non-glaucoma individuals without serious diseases, the results will be presented not only in visual, but also in statistical resolution.
Statistical analysis was added in Material and Methods in manuscript.
2.1 Statistical analysis
Shapiro-Wilk test was used for testing of data normality. Mann-Whitney U test and Unpaired one-sided t-test were used to analyze differences between groups CTRL and PGOU for synchronous spectrum Δλ = 110 nm and derivative synchronous spectrum, respectively, p-Value < 0.05 was considered statistically significant.
Statistical analysis was added in Results:
Table 2. Results of statistical analysis of patients POAG and healthy subjects (CTRL)
|
|
Axis |
Data |
Average |
STD |
Test |
p-value |
Result |
|
SP |
x: ("-1w"/"+1w")*F |
2d_SP_CTRL |
1.23 |
0.34 |
t test |
0.000207 |
significant |
|
2d_SP_POAG |
2.39 |
0.62 |
|||||
|
y: ("-1"/"+1")/("-2"/"+2") |
2d_SP_CTRL |
0.25 |
0.06 |
t test |
< 0.00001 |
significant |
|
|
2d_SP_POAG |
0.44 |
0.05 |
|||||
|
Δλ = 110 |
x: ("-1w"/"+1w")*F |
2d_110_CTRL |
2.55 |
0.61 |
t test |
0.84 |
insignificant |
|
2d_110_POAG |
2.6 |
0.54 |
|||||
|
y: ("-1"/"+1")/("-2"/"+2") |
2d_110_CTRL |
0.79 |
0.07 |
Mann-Whitney U test |
0.0083 |
significant |
|
|
2d_110_POAG |
0.93 |
0.11 |
|||||
|
|
|
|
|
|
|
|
|
Specific comments
Point 7) Specific comment 1) Lines 86 – 88: These are the journal guidelines and need to be deleted.
Response 3 (point 7): The journal guidelines were deleted.
Point 8) Specific comment 2) Line 99: “Both,” – remove comma
Response 3 (point 8): Comma was removed in line 99: “Both synchronous spectra…….
Point 9) Specific comment 3) Lines 126-127: “The peaks “-1”; “+1” and “-2” show the largest differences in the shape…” – Do the Authors mean “-2” or “+2”? My question is in the sense that by visual inspection it seems that the peak “+2” shows a larger difference than the peak “-2”.
Response 3 (point 9 There are larger shape differences in the "-2" peak than in the "+2" peak (Figure 5, detail), but they are still small
Figure 5 - detail
Point 10) Line 144: “The ratio (“1/+1”)/(-2/+2) expresses” – Add “-“ before the 1.
Response 3 (point 10): The sentence was changed as suggested: (“1“/“+1”)/(“-2“/“+2“)
Point 11) Legend of Figure 6 must be complemented with the information concerning the mathematical process behind the data presented so that the reader does not need to check the manuscript to understand this figure.
Response 3 (point 11): We added more details in caption of Figure 6 as you suggested:
Figure 6. Comparison of the mathematical processing of two different ways of using 3D derivative fluorescence analysis to distinguish between glaucoma patients (POAG) and healthy subjects with normal intraocular pressure (CTRL). Graphical differentiation between tear fluid of glaucoma patients and healthy subjects with normal intraocular pressure.
- Mathematically processed the 2nd derivative synchronous profile (SP) of glaucoma patients and healthy subjects tear fluid. CTRL_AVG. B. Mathematically processed the 2nd derivative synchronous spectrum Δλ= 110 nm of glaucoma patients and healthy subjects tear fluid.
The 2nd derivative of both synchronous profile and synchronous spectrum Δλ= 110 nm were mathematically processed, and the result was plotted on a graph.
The ratio (“1“/“+1”)/(“-2“/“+2“) expresses the relationship between the two fluorescence peaks and characterizes the whole synchronous profile. In contrast, the ratio (“-1w”/”+1w”) represents only the first fluorescence center where the differences between the CTRL and POAG groups were greatest.
The large point represents the average (AVG) of the results in the group of POAG and CTRL
The same patients (POAG) and healthy subjects (CTRL) were used for the analyses of the synchronous profile and Δλ= 110 nm synchronous spectrum.
The procedure for calculating the value given as (“-1w/+1w)*F is not explained in detail, it is being considered for patenting.
Point 12) Some proofreading and English correction is necessary.
Response 3 (point 12): Proofreading and English correction was carried out.
We would like to thank the reviewer for his stimulating comments.
With best regards,
authors
Assoc. prof. Katarína Dubayová, PhD., RNDr., Kristína Krajčíková, PhD., prof. Mária Mareková, CSc and Assoc prof. Vladimíra Tomečková, PhD. (Corresponding author)

Round 2
Reviewer 2 Report
No more comments.